# Vascular Plants Flora of Mire Ecosystem of the Bolshoy Shantar Island (the Far East of Russia)

**DOI:** 10.3390/plants11060723

**Published:** 2022-03-08

**Authors:** Viktoriya A. Kuptsova, Lyubov A. Antonova, Vladimir V. Chakov

**Affiliations:** Khabarovsk Federal Research Centre FEB RAS, Institute of Water and Ecology Problems FEB RAS, 56 Dikopoltseva, 680021 Khabarovsk, Russia; levczik@yandex.ru (L.A.A.); chakov@ivep.as.khb.ru (V.V.C.)

**Keywords:** Bolshoy Shantar, mire, ecosystem, vascular, plants, island, flora, taiga, zone

## Abstract

This article presents the findings of the authors’ study of the mire ecosystem vascular plants of the island of Bolshoy Shantar, which is the largest island in the Shantar archipelago. Bolshoy Shantar Island is an insular ecosystem, the study of which can provide insight into the natural “polygons” of evolution at work. The botanical research was conducted through the application of traditional techniques of floristic and geobotanical studies. The material for this article was drawn from 73 floristic and 54 geobotanical descriptions made between 2016–2018 in the north-eastern part of Bolshoy Shantar Island on four mire massifs associated with various hypsometric surfaces. The findings of this study indicate that the flora of vascular plants of the mires of Bolshoy Shantar Island reflect the peculiarities of a regional mire type that formed in the insular conditions of the Pacific. The species richness of the vascular flora of the island’s mire ecosystems is evidenced by a total species count of 158, composed of 109 genera and 48 families, which accounts for more than one quarter (26.3%) of the Shantar archipelago’s flora. Over half of these species (63.7%) form the core of the mire flora.

## 1. Introduction

The *Shantar* archipelago is the largest in the marine territory of Khabarovsk Krai, with a total length of over 2000 km. It is located in the south-west part of the Okhotsk Sea between 54° N and 55° N and 136° E and 139° E. The archipelago comprises 15 islands, the largest of which is Bolshoy Shantar Island. The islands were formed as a result of fragmentation of the mainland margins and their subsidence below sea level at a Middle–Late Holocene Boundary 9000–10,000 years ago [1,2]. The proximity of the Shantar archipelago to the mainland (20–130 km), and its recent separation, has led to a low level of the island isolation effect [3], which is confirmed by the absence of island endemism in plants [4]. The landscape structure of the islands is close to the continental landscapes. However, the islands are unique natural systems. The peculiarity of the terrestrial and aquatic ecosystems of the Shantar Islands lies not in the species richness and the endemism of its representatives, but in the existence of the biota of various origins and ecology, which forms ecosystems of complex composition and structure in the limited islands area [5].

Shantar Islands are a specially protected natural area of the Shantar Islands National Park since 2013. This protected area is an important link in the ecological network of international marine protected areas. According to the functional zoning of the national park [6], the ecosystems surveyed in the basins of the Maly Argulad, Bolshoy Argulad, Tundrobaya rivers, and the eastern part of the Bolshoe Lake catchment area are key botanical areas and are included in the strict protection zone or the protected island territory.

The unique nature of the areas is closely related to the fact that the islands are located in the coldest part of the Sea of Okhotsk, where drifting ice is concentrated in the summer. Regional differences in the islands’ climatic rhythms have depended on the presence of ice. It was expressed in the contrast and metachronism of paleoclimatic events. Thus, here, contrary to other areas of the Okhotsk region, the end of the Late Glacial period (Young Dryas) was very cold, and the early and middle Holocene optima did not occur or was limited [2]. At present, the weather on the islands is also more severe than on the mainland coast. Strong winds, excessive moisture, lower average annual temperatures, and a delay in the beginning of the growing season by one to two months compared to the mainland are important factors in the formation of the structure of vegetation cover.

In this regard, a comparative analysis of the flora of insular and marginal continental landscapes is of fundamental and applied importance.

The strategy of this scientific research was to establish and compare the zonal regional and local patterns of mire vegetation of Bolshoy Shantar and marginal continental ecotone bio-geosystems. A comparative analysis of the individual cenofloras of the principal vegetation groups will be highly informative and will show deep patterns in the composition and structure of the flora of the territories compared.

The data obtained will complement the information on sub-latitudinal changes in landscape relations in the Pacific Mega Ecotone of Northern Eurasia. They may also be used to develop scientific and methodological issues of sustainability of natural landscapes with the impact from natural and anthropogenic factors.

This article presents the outcomes of a study on the vascular plant flora of Bolshoy Shantar. The second stage of the work will involve a comparative analysis with the flora of the continental mire ecosystems of the coast of the Sea of Okhotsk. This will be undertaken following similar studies in the adjacent continental territory. At present, this analysis is not possible because of the lack of data. No special studies have been conducted on the flora of vascular plants in the mire ecosystems of the Khabarovsk Krai. Information on species that grow in mires can be obtained from studies of flora and vegetation in northern Khabarovsk Krai [4,7,8,9,10]. They are shown in a general overview of species in large areas (Uda River basin, western coast of the Sea of Okhotsk, etc.), or incomplete lists of vegetation descriptions, and are insufficient to obtain reliable comparison results.

The first recorded data available on the Shantar archipelago’s vegetation was published by A.F. Middendorf who visited the islands in 1844 [11]. In the proceeding decades, the body of research on the islands’ flora has grown significantly [4,12,13,14]. This body of research has established that the islands’ flora is represented by 600 species of vascular plants from 282 genera belonging to 85 families and forming 24.4% of the Khabarovsk Krai flora. Specific observations have included the identification of rare protected plant species and detailed features of coastal-marine flora [15,16,17]. This current study of the archipelago’s mire ecosystem vascular plants is the first of its kind.

According to Kolesnikov’s [18] geobotanical zonation scheme, the Shantar archipelago is zoned as a South-Okhotsk dark coniferous forest subregion of the mountain-coastal Ayano-Shantar district. The forest vegetation of the islands is dominated by spruce and larch forests. The larch forests were formed as a result of fires associated with human economic activity in the late 19th and early 20th centuries [12]. The major areas of the forest belt contain intrazonal vegetation formations of rocks, mires and coastal meadows [4]. The mire vegetation is concentrated on the Priozernaya lowland of Bolshoy Shantar Island, and the southern part of Maly Shantar Island. The mires concentrated within the valley part of the Tundrovaya and Olenya rivers, as well as on the Priozernaya lowland of the Bolshoy Shantar Island and the southern part of the Maly Shantar. Following to mire classification of the southern part of the Russian Far East developed by Yu.S. Prozorov [7], oligotrophic sphagnum phytocenoses dominated by *Sphagnum divinum* Flatberg & K. Hassel, *S. fuscum* (Schimp.) H.Klinggr, and *S. capillifolium* (Ehrh.) Hedw. are mainly confined to the flat areas of the valleys of Tundrovaya and Olen’ya river and their tributaries with hypsometric elevations from 20 to 90 m. The lakeside lowland with altitudes below 20 m is characterized exclusively by herbaceous fen communities dominated by sedges and reed grass (*Carex cryptocarpa* C.A.Mey., *C. rariflora* Smith, *Calamagrostis purpurea* (Trin.) Trin.).

The overall aim of this current study was to determine zonal regional and local regularities in the organization of the vegetation cover of mires of ecotone bio-geosystems of landscapes of the regressive series of Bolshoy Shantar Island. Moreoever, the study aimed to gain insight into the species richness of the mire ecosystems of Bolshoy Shantar Island, to identify the core mire flora and the main groups of mire ecosystems, as well as the confinement to them of both typical and random species based on their activity. Floristic analysis was also carried out to identify the specific features of the studied mire ecosystems.

## 2. Results

The results of this study established that the vascular plant flora of the Bolshoy Shantar Island mire ecosystems total 158 species, comprising 109 genera and 48 families, and comprising more than one quarter (26.3%) of the Shantar archipelago flora [5].

### 2.1. Species Categorization

In order to highlight the specific features of the mire flora of the ecosystem studied, our research identified species that are exclusively or predominantly characteristic of mires and represent a core of the mire flora, or coenotic complex [19]. To differentiate the species, we estimated their “constancy” in the mire biotopes by applying the adapted J. Braun-Blanquet scale [20], which is suggested for determining the fidelity of species to syntaxons of vegetation classification.

The scale includes five categories:

I–species appearing in mires rarely or randomly;

II–non-mire (indifferent) species capable of growing in mires in accordance with their ecological preferences;

III–species characterized by a high consistency in mires but able to grow in forests, wet meadows and along waterbody banks;

IV–species preferring mire biotopes and often found therein but sometimes growing in other types of habitats as well;

V–species characteristic of mire biotopes only. The fidelity for each species is determined with due consideration to the peculiarities of distribution in the mires of the Russian Far East.

The smallest number of species identified are those found on mires randomly or rarely (Category I) and include *Polemonium schmidtii* Klokov, *Aconitum umbrosum* (Korsh.) Kom., *Coptis trifolia* (L.) Salisb., *Solidago spiraeifolia* Fisch. ex Herder., *Euphrasia maximowiczii* Wettst., and *E. ajanensis* Vorosch. Category I species comprise only 3.8% of the flora volume of Bolshoy Shantar Island’s mire ecosystems. The group of indifferent species (Category II) is the most numerous. We identified 51 species in this category, found predominantly outside the mires, comprising 32.5% of the island’s flora volume. Category II species identified include *Equisetum sylvaticum* L., *Pyrola rotundifolia* L., *Ranunculus repens* L., *Urtica angustifolia* Fisch. ex Hornem., *Chamaenerion angustifolium* (L.) Scop., *Gymnadenia conopsea* (L.) R.Br., *Aegopodium alpestre* Ledeb., *Maianthemum bifolium* (L.) F. W. Schmidt, *Lysichiton camtschatcense* (L.) Schott, *Arctous alpina* (L.) Niedenzu, *Rhododendron dauricum* L., *Anemonoides debilis* (Fisch. ex Turcz.) Holub and *Filipendula palmata* (Pall.) Maxim. The species mentioned are forest, meadow, coastal-aquatic and aquatic. Key species identified in Category III include *Epilobium palustre* L., *Lathyrus pilosus* Cham., *Veratrum lobelianum* Bernh., *Iris setosa* Pall. ex Link, *Stachys aspera* Michx. and *Ostericum maximowiczii* (Fr. Schmidt) Kitag. We identified 35 such species, which corresponds to 22.3% of Bolshoy Shantar Island’s mire ecosystem flora. Species in Categories IV and V are considered as “faithful” to mire biotopes and comprise the core of the mire flora of Bolshoy Shantar Island. Our research identified 65 species (40.1%) from 49 genera and 28 families in these two categories. Key species identified in Category IV include *Carex globularis* L., *Ledum palustre* L., *Parnassia palustris* L., *Caltha palustris* L. and *Vaccinium uliginosum* L. Category IV comprised 39 species, corresponding to 24.8% of the total species of Bolshoy Shantar Island’s mire ecosystem. Key species identified in Category V include *Andromeda polifolia* L., *Chamaedaphne calyculata* (L.) Moench, *Oxycoccus palustris* Pers., *O. microcarpus* Turcz. ex Rupr., *Rubus chamaemorus* L., *Drosera rotundifolia* L., *Carex limosa* L., *C. pauciflora* Lightf., *Eriophorum vaginatum* L., *Rhynchospora alba* (L.) Vahl and *Menyanthes trifoliata* L. We identified 26 such species, which corresponds to 16.6% of Bolshoy Shantar Island’s mire ecosystem floral composition.

### 2.2. Species Categorization—Regional Comparison

The adaptation of the J. Braun-Blanquet scale to identify the core species of mire flora, as detailed in Section 2.1, has also been applied in other regions of Russia [21,22,23]. No similar studies have been conducted on the flora of vascular plants in the mires of the Russian Far East. When comparing the structures of Bolshoy Shantar Island’s mire floras with the western regions of Russia, we observed the specific features of the flora studied, which are indicated by a high number of the species forming its core. In many regional floras, the mire flora core includes not only Category IV and V species, but also Category III, which can comprise one-third of the flora of mire ecosystems. Thus, the mire flora core of the mire ecosystems comprises 32.8% of the Central Russian Upland [24] and 45.2% in Chelyabinsk Oblast [25] of total species. When identifying the core of the mire flora of Bolshoy Shantar Island, species of Categories IV and V have only been included. In this instance, the share of species forming the floracoenotic complex of mire flora is 41.4%. However, when including Category III species, the core of the Bolshoy Shantar Island flora amounts to 63.7%. Such a high proportion of species comprising the core of the mire flora is a result of the specific, severe, natural and climatic conditions of the insular ecosystem of the Okhotsk Sea.

Another distinct feature of the vascular plant flora of the mire ecosystems studied is the very insignificant (3.8%) role of random species (Category I). In comparison, in the intact mires of the Central Russian Upland, the random species amount to 29.5%, and the floral composition of transformed mire ecosystems is over 60% [24]. The group of indifferent species (Category II) is the most numerous of the floras being compared and amounts to approximately one-third of the species richness.

A literature review of mire vegetation in the adjacent mainland region allowed the inclusion of only 73 mainland species in the Bolshoy ecosystems. This confirms the need for special studies of the flora of these ecosystems [7,9]. When describing the mire vegetation, the authors of these works usually considered only the species that form the core mire flora. Characterized by high sustainability in mires, species capable of growing in other ecotopes that are indifferent to mire were not considered. However, these species groups are indicators of the degree of transformation or specificity of abiotic components.

To understand the specific features of the vascular plant flora of the mire ecosystems of Bolshoy Shantar Island, our research analysed its taxonomic structure. The entire spectrum is usually not used, but its “head part” (head spectrum) represented by a set of up to 10 leading families is considered for this purpose. Table 1 details the head part of the taxonomic spectrum of the flora of the Shantar Archipelago, mire ecosystems of Bolshoy Shantar Island and the mire flora core of Bolshoy Shantar Island and includes the rank in order of decreasing their percentage share of species in a family.

The head part of the family spectrum of the mire ecosystem flora of Bolshoy Shantar Island and the Shantar Archipelago flora is composed of the same families, except for the *Salicacea* family. The leading positions in the mire flora studied are occupied by two families, the composition of which includes species that form of mire communities—*Cyperaceae* and *Ericaceae*. In the mire flora core, *Cyperaceae* ranks first and comprises a third of species (33.4%). This is a natural phenomenon; the high ranking of *Cyperaceae* is characteristic of mire flora. As in many regional mire floras in the taiga zone (Taiga zone, also called boreal forest region of North America and Eurasia, are broad belts of vegetation that span their respective continents from Atlantic to Pacific coasts), the number of species *Cyperaceae* is substantially higher than that of other families.

The strong role of *Ericaceae* is also characteristic of the floracoenotic complexes of the taiga zone [22,25,26]. This family ranks eighth in terms of the number of species in the Shantar Archipelago mire flora and it ranks second in the core of mire flora.

### 2.3. Taxonomic Structure

Just as in floracoenotic complexes of taiga zone mires [21,22,23], in the head spectrum of the mire flora of Bolshoy Shantar Island, the rank of *Ranunculaceae* and *Asteraceae* is not significant, with representatives of the latter not entering the core of the studied mire flora at all.

Thus, the taxonomic structure of the mire flora of the studied area is not fundamentally different from those in the regions of the taiga zone. This indicates a similarity of mire biotope conditions in these climatic conditions, which determines a homogeneity of composition and a characteristic structure of mire flora.

Specific features of the taxonomic structure are noted in the core of the mire flora of Bolshoy Shantar Island. They are expressed by a high concentration of species in the head spectrum (72.3%), the appearance of the *Juncaginaceae*, *Juncaceae*, *Polygonaceae* and *Droseraceae* families and an absence of the *Salicaceae* and *Orchidaceae* families in the head spectrum. In the composition of the first triad (the first three family ranks) of the mire flora core, the *Poaceae* family substitutes the *Rosaceae* family, which usually occupies a leading position in the mire flora core [25].

In the genus-species spectrum and in all flora of the Shantar Archipelago, we observed a high preponderance of the *Carex* genus relative to other genera. In the Bolshoy Shantar Island mires, the *Carex* genus is represented by 15 species, while the biggest part of genera (92.7%) has one or two species and only seven genera include three or four species (*Equisetum*, *Poa*, *Calamagrostis*, *Bistorta*, *Thalictrum*, *Pedicularis*).

Geographical analysis indicates that the island mire flora and its core are characterized by a prevalence of circumpolar and circumboreal (30% and 40%, respectively) species (*Carex rotundata* Wahlenb., *Rubus arcticus* L., *Comarum palustre* L., *Baeothryon cespitosum* (L.) A. Dietr., *B. alpinum* (L.) Egor., *Eriophorum russiolum* Fries, *E. vaginatum*, *Drosera anglica* Huds., *D. rotundifolia*, *Equisetum fluviatile* L., *E. pratense* L., *E. sylvaticum* L., *Andromeda polifolia*, *Chamaedaphne calyculata*, *Oxycoccus microcarpus*, *O. palustris*, *Phyllodoce caerulea* (L.) Bab., *Rhodococcum vitis-idaea* (L.) Avror., *Vaccinium uliginosum* and others). Moreover, the mire flora and its core of the island have a high percentage of the Eurasian-North American (17% and 21%, respectively) and Eurasian (11% and 12%, respectively) species. In total, these geo-elements amount, in total, to 58% and 73%, respectively. A key feature for both floras of cold and moderately cold belts of the northern hemisphere and, in particular, for mire ecosystems is the significant proportion of species with a vast holarctic areal [22,27]. A significant percentage of the mire flora and its core in the studied territory comprises Siberian-Russian Far Eastern (16% and 9%, respectively), Siberian-North American (8% each) and Russian Far Eastern-North American (9% and 5%, respectively) species. This can be explained by the geographical location of the area of study. In general, a smaller presence of the Siberian species of plants is peculiar to the coast of the Okhotsk Sea [17].

The share of the Russian Far Eastern species in the mire flora and its core amounts to 9% and 5%, respectively (*Ostericum maximowiczii*, *Cirsium schantarense* Trautv. et Mey., *Lonicera chamissoi* Bunge ex P. Kir., *Sagina maxima* A. Gray, *Carex middendorfii* Fr. Schmidt, *Swertia tetrapetala* Pall., *Myrica tomentosa* (DC.) Aschers. et Graebn., *Picea ajanensis* (Lindl. et Gord.) Fisch. ex Carr., *Aconitum umbrosum*, *Anemonoides debilis*, *Rubia jesoensis* (Miq.) Miyabe et Miyake, *Euphrasia ajanensis*, *E. maximowiczii* and others). This is substantially lower than in the flora of Khabarovsk Krai, where the share of vascular plants amounts to 32.2% of total plant species [8]. This difference can be correlated to the fact that in the Russian Far Eastern group of species, a prevalent position is occupied by a nemoral complex, of which few are present on the Shantar Archipelago, also distributed over the north-east part of China, the Korean peninsula and the Japanese islands.

Our study notes that the flora core, as detailed in Figure 1, has less than half the share of plant species, with the narrow range including the Siberian-Russian Far Eastern, the Russian Far Eastern, the Russian Far Eastern-North American, and an increased share of species with wide ranges.

Our geographical analysis of flora by the composition of longitudinal groups indicates a prevalence of boreal species (72% of the identified vascular plants), which are widely spread in the taiga zone in general, and are observed in both coniferous forests and mires. The absence of nemoral species in the mire flora, which are only present in small numbers in the floras of the Shantar Archipelago [4], can be attributed to the ecotopic conditions of the mires.

The specificity of the species composition of the Shantar Archipelago is a result of a group of plants that are genetically related to those of the Pacific coast and were formed on its littoral a long time ago. Examples of this include *Leymus mollis* (Trin.) Hara, *Arctopoa eminens* (C. Presl) Probat., *Rosa rugosa* Thunb. and *Myrica tomentosa*. A series of species with oceanic origin are linked to this group, the ranges of which typically cover the continental coast of the Okhotsk Sea, Sakhalin Islands, the Kuril Islands and Japan (*Lysichiton camtschatcense*, *Lathyrus japonicas* Willd., *Rubia jesoensis* and others). Montane species also play an important role in the mire communities and are widespread in the Arctic and on the montane ecosystems of the Eurasian continent (*Empetrum sibiricum* V. Vassil., *Arctous alpina* and others).

These mire ecosystems differ from the same coastal and continental ecosystems in a greater prevalence of organogenic cryogenic landforms and their size. As a result, remnant permafrost landforms can reach up to 100 m in diameter in the valley of the Amur River, but in the island mires, they barely reach 10 m in diameter, though similar heights of 1.8–2.5 m. The former is characterized by a predominance of woody vegetation (larch, aspen and birch), and the latter are mostly covered by shrubs and dwarf shrubs (*Pinus pumila* (Pall.) Regel, *Empetrum spp.*, *Rubus chamaemorus*). In most cases, the landforms are in contact with the thermokarst depressions, which appear as hollow, often water-filled. Vascular plant species (*Baeothryon alpinum*, *Baeothryon cespitosum*, *Menyanthes trifoliata*, *Calla palustris* L., *Sanguisorba parviflora* (Maxim.) Takeda) that are typical of eutrophic or eutrophic-mesotrophic conditions most often grow in the hollows.

The contrast of the vegetation coverage of the mire ecosystems of the Bolshoy Shantar is most apparent in the regressive complexes, the nature of which is due to the features of the coastal marine climate against the background of global warming over the last few decades. This has significantly increased the intensity of the thermokarst processes that occur throughout the permafrost zone of the northern hemisphere [28,29,30].

### 2.4. Topological-Ecological Classification

A topological-ecological classification of the mire plants of the territory studied was conducted to determine the role of vascular plants in forming mire ecosystems. The classification was divided into three steps: association; groups of associations; and class.

Based on the dominant species and ecological-coenotic groups of species, this study separated the united associations into groups of associations determined by habitats and proximity of communities’ species composition. When separating the associations, we considered the location of mire plots and communities in the ecological profile, as well as correlation with microrelief. In Table 2 the name of the association is allocated according to dominant or characteristic (diagnostical) species. Class separation was determined through a combination of three criteria: ecological (type of water-mineral nutrition and trophicity of habitats), phytocoenotic, and topological (ground-water level within habitats and correlation of syntaxons with elements of microrelief).

### 2.5. Groups of Association

In each of the five groups of associations, our study established a species richness, with the “activity” of each species determined based on its constancy (occurrence frequency) and abundance (projective cover). In this regard, by using the activity of higher vascular plants, which is an integral index of occurrence and abundance, we can assess the stability of mire phytocenoses under changing natural conditions and predict the dynamics of their structural changes [31,32]. This data is presented in Table 3, Table 4, Table 5, Table 6 and Table 7. To characterize the quantitative participation of species in composing the plant associations (with small corrections of the point values +, 1, 2), we applied the Braun-Blanquet scale [20] of species cover classes: +–<1%; 1–1–5%; 2–5–25%; 3–25–50%; 4–50–75%; 5–projective cover of species over 75%.

The constancy of species is defined as the quantitative characteristic of participation of species in composing a plant community (percentage of sampled areas where the species is found, from the total number of examined areas). Our study allocated five classes of constancy (with 20% class volume):I class—species occurred on more than 20% of areas;II class—species occurred on 20–40% of areas;III class—species occurred on 40–60% of areas;IV class—species occurred on 60 to 80% of areas; andV class—species occurred on >80% of areas.

In total, 42 species are present in the sedge group of associations, of which 18 (42.9%) are active, including two species of shrubs (*Myrica tomentosa*, *Salix fuscescens* Anderss.). The most active species are sedges (*Carex middendorfii*, *C. cryptocarpa* C. A. Mey., *C. rariflora*, *C. cinerea* Poll.; *C. globularis*, *Carex loliacea* L.), purple reedgrass (C*alamagrostis purpurea*) and the ericaceous dwarf shrub *Empetrum sibiricum* (Figure 2). A high occurrence at the low projective cover (not above 5%) of mire species is noted: *Drosera rotundifolia*, *Parnassia palustris*, *Oxycoccus palustris*, *Comarum palustre*; meadow-mire: *Sanguisorba tenuifolia*, *Iris setosa*; and coastal: *Arctanthemum arcticum* (L.) Tzvel., *Chamaepericlymenum suecicum* (L.) Aschers. et Graebn.

The total number of species described for the reedgrass group of associations is 71, 10 of which are active (14.1%) (Table 4). The dominant species are: *Calamagrostis purpurea*; subdominants–sedges: *Carex cryptocarpa*, *C. middendorfii*, halophilous *Leymus mollis*, *Senecio cannabifolius* Less. and *Comarum palustre* (Figure 3). *Other species (Ranunculus repens* L., *Sanguisorba tenuifolia*, *Iris setosa*, *Fimbripetalum radians* (L.) Ikonn.) occur frequently, but have projective cover not above 5%.

Some 37 species are present in the groups of associations of coastal marshes, of which 18 (48.6%) are active species, including two species of shrubs (*Myrica tomentosa*, *Salix fuscescens)* (Table 5). The dominants of herbal layer are *Triglochin palustre* L., *Potentilla pacifica* Howell, *Arctopoa eminens*, *Carex loliacea*, *Lathyrus pilosus*, *Juncus haenkei* E. Mey., *Calamagrostis purpurea*, *Leymus mollis* and *Calamagrostis langsdorffii* (Link) Trin. (Figure 4). A high occurrence frequency at abundance below 5% is found with *Arctanthemum arcticum*, *Stellaria humifusa* Rottb., *Sanguisorba tenuifolia* Fisch. ex Link, *Geranium erianthum* DC., *Chamaepericlymenum suecicum*, *Allium maximowiczii* Regel and *Iris setosa.*

Of the total of 61 species in the composition of shrub associations, 16 (26.2%) are active (Table 6). Half of these are shrubs and dwarf shrubs, with the highest activity characterized by *Myrica tomentosa*, *Salix fuscescens*, *Vaccinium uliginosum* (Figure 5). Highly active species are sedges, *Carex middendorfii*, *C. cryptocarpa*, *C. globularis*, mire species of *Andromeda polifolia* and *Parnassia palustris* and Pacific species, *Chamaepericlymenum suecicum*. *Sanguisorba tenuifolia* and *Iris setosa* have high constancy, with the projecting cover of these species only 1–5%.

The total composition of the sphagnum group of associations is 56 species, of which 16 (28.6%) are active (Table 7). The group of active species is dominated by shrubs and dwarf shrubs, which total 10 (62.5%) species (Figure 6). High activity is also noted by representatives of the *Cyperaceae* family.

Thus, the largest species richness belongs to the reedgrass (71 species) and shrubs (61 species) groups of associations. The smallest number of species are represented by the coastal marshes (37 species) and sedge (42 species) group of associations. Within these two groups of association, the share of active species is much higher in the sedge (42.9%) associations and coastal marshes (48.6%), while in reed-grass and shrubs it is much lower and amounts to 14.1% and 26.2%, respectively. More detail on the ratios within the five groups of association is outlined in Figure 7.

The main coenoforming species of mire communities are the active species with high indicators of projected cover and occurrence. B. A. Yurtsev [33] (p. 5) believed that “the eco-biological properties of active species correspond to a general landscape-climatic situation of the given territory, and it finds its expression in an elevated number of such species, significant width of their ecological amplitude, more spaced distribution of them on the territory, that is in a more intensive capturing of the given landscape by these species”.

In total, there are 41 such species counted in all groups of plants associations, which amounts to 25.8% of the studied flora. Among them, there are species common in two or four groups of associations (*Calamagrostis purpurea*, *Carex cryptocarpa*, *C. globularis*, *Chamaepericlymenum suecicum*, *Salix fuscescens*, *Myrica tomentosa* and others). Of particular note are the species where activity is first determined by a high occurrence frequency at a low projected cover. Such species have been identified on each of the 54 described sample areas, and include *Iris setosa*, *Sanguisorba tenuifolia* and *Parnassia palustris.*

Some species are active in a single group association only. Most of these species are in the communities of coastal marshes (nine species)—*Allium maximowiczii*, *Stellaria humifusa*, *Triglochin palustre* L., *Potentilla pacifica*, *Lathyrus pilosus*, *Juncus haenkei*, *Carex loliacea* and *Arctopoa eminens*; and in the group of sphagnum associations (six species)***—****Betula exilis*, *Carex rotundata*, *Chamaedaphne calyculata*, *Eriophorum russiolum*, *Rubus chamaemorus* and *Oxycoccus microcarpus.* There are notably fewer of the remaining species in the groups of associations: in the sedge, one species (*Carex rariflora*); in the reed-grass, three species (*Fimbripetalum radians*, *Ranunculus repens* and *Senecio cannabifolia*); and in the shrub, one species (*Spiraea beauverdiana* Schneid.).

Shrubs and dwarf shrubs play a significant role in forming the vegetation cover of the studied mire communities, with the exception of the reed-grass group of associations. Shrubs comprise 50% of all active species (*Myrica tomentosa*, *Salix fuscescens*, *Empetrum sibiricum* and others) and 62.5% of the sphagnum group; and a high diversity apart from *Myrica tomentosa* and *Salix fuscescens* is ensured by ericoid shrubs and dwarf shrubs (*Andromeda polifolia*, *Oxycoccus microcarpus*, *O. palustris*, *Vaccinium uliginosum*, *Empetrum sibiricum* and *Chamaedaphne calyculata).* In the sedge group and the coastal marshes, the share of active and dwarf shrubs is lower, at 16.6% and 11.1%, respectively. The highest parameters of activity belong to *Myrica tomentosa*, *Salix fuscescens* and *Empetrum sibiricum* dwarf shrub.

## 3. Discussion

Our study indicates that the flora of vascular plants of the mire ecosystems of Bolshoy Shantar Island reflect peculiarities of a regional mire type that formed in the insular conditions of the Pacific. The species richness of vascular plants of the island’s mire ecosystems is evidenced by a total species count of 158, composed of 109 genera and 48 families, which account for more than one quarter (26.3%) of the Shantar archipelago’s flora. Over half of these species (63.7%) form the core of the mire flora.

In terms of the composition of the most abundant families (leading families), the mire flora is almost analogous to the flora of the Shantar Archipelago, while first place in terms of species abundance is occupied by the *Cyperaceae* family, which is also characteristic of the mire floras of the taiga zone in general. The specific features of the taxonomic structure are shown in the core of the mire ecosystems’ flora and expressed as a high number of species in the head spectrum of the core of mire flora (72.3%), and occurrence in the head spectrum of the core of species of *Juncaginaceae*, *Juncaceae*, *Polygonaceae* and *Droseraceae*, and absence of the families *Salicaceae* and *Orchidaceae*. In the genus-species spectrum, we observed a high preponderance of species in the genus *Carex* relative to other genera, while most of the genera are represented by one or two species.

In the spectrum of geo-elements and types of ranges of the Bolshoy Shantar Island’s mire flora, a high proportion are boreal and holarctic species, which is characteristic of the mire floras of the taiga zone in general [22,23]. The specificity of geographical structure is ensured by the Russian Far Eastern and Russian Far Eastern-North American groups of plants, genetically related to those of the Pacific coast, and also by montane species widespread in the Arctic and in montane ecosystems of the Eurasian continent.

The vegetation of the mires is represented by three classes, five groups of associations and 17 associations. The species richness in the groups of associations varies from 37 in the coastal marshes to 71 species in the reed-grass group of association. The share of active species ranges from 14.1% to 48.6%. In the composition of active species, the major role belongs to shrubs and dwarf shrubs.

The mire ecosystems of the island are different from the coastal and continental ecosystems in that they have a large distribution and size of organogenic cryogenic landforms. They are much smaller on Bolshoy Shantar than on the mainland. The tree layer here is mainly represented by hypoarctic shrubs and shrubs and looks like tundra. The contrast of the vegetation of the insular ecosystems considered is most visibly reflected in regressive mire complexes, the nature of which is due to the characteristics of the coastal marine climate against the background of its warming in recent decades.

Our research has confirmed that there are 158 species of vascular plants in the flora of vascular plants in Bolshoy Shantar mires. Analysis of literature of vegetation in adjacent mainland areas shows that only 73 species in similar mainland mire ecosystems are present on the Bolshoy Shantar mires, confirming the need for special studies on the flora of mires. When describing the mire vegetation, the authors of these papers usually take into account only the species that form the core of the mire flora. Species characterized by high persistence in mires but capable of growing in other ecotopes, species that are indifferent to mire, and random species were not taken into account, but it is these species groups that are indicators of the degree of transformation or the specificity of abiotic factors.

Nevertheless, preliminary comparative analysis can already reveal the specific features of the Bolshoy Shantar mire flora. It is defined by a group of plants that are genetically related to the Pacific coast. There are *Rosa rugosa*, *Myrica tomentosa*, *Leymus mollis*, *Arctopoa eminens*, and others. This group includes species of oceanic origin (*Lysichiton camtschatcense*, *Rubia jesoensis*, *Lathyrus japonicus*, etc.) that characterize the the mainland coast of the Sea of Okhotsk, Sakhalin, the Kuriles and Japan. Mountain species (*Empetrum sibiricum*, *Arctous alpina*, etc.) widely distributed in the Arctic and Eurasian continental mountains also play an important role in the formation of mire communities.

Therefore, the island effect that normally appears in flora depletion may not be represented by the flora of the mire ecosystems of Bolshoy Shantar, and conversely, the flora structure is complicated by the diversity of the floristic composition due to arcto-montane and Pacific species. Due to the effects of the sea and the harsh climate, the flora structure of the island’s mires has become more complex than that of the mires of the continental margins. At present, such comparative analysis is not possible due to the lack of complete data. This study aims to provide direction for future research in other parts of the Russian Far East.

## 4. Materials and Methods

To study the flora and vegetation of the mires of the Bolshoy Island, a field description of flora and vegetation was carried out at its north-eastern end from 2016 to 2018. The methodical basis of the research consists of traditional techniques of floristic and geobotanical studies [34].

To analyse mire vegetation, we used the data of 73 floristic and 54 geobotanical descriptions (relevees) on four mire massifs associated with various hypsometric surfaces (see Figure 8): estuary of Lake Bolshoe (2.0–5.0 m.a.s.l.); northwestern coastal part of the Lake Bolshoe (2.0–35.0 m.a.s.l.); interstream area of the Argulad and upper course of the Tundrovaya River (75.0–90.0 m.a.s.l.); and the Levyi Argulad (30.0–50.0 m.a.s.l.).

On each of these four mire massifs, within the natural borders of the mire communities, we laid ecological profiles, the descriptions of which have been included in this study. Geobotanical descriptions were performed following the standard method [35]. The descriptions were carried out on sample plots of 100 m^2^, less commonly 400 m^2^. They indicated forest stand composition and density (if applicable), total projective coverage (%) for the grass, herb-shrub and moss layers, as well as the projective cover for each species. Floristic descriptions were made on the profile outside the temporary sample plot. A total of 127 descriptions were inputted into an electronic database with MS Excel software. It was used to treat the descriptions and construct a topological-ecological classification of mire vegetation in the study area [23]. The authors identified the species richness of associations [36] and activity (an integral indicator of the occurrence and abundance of a vascular plant species [31,32].

The list of flora was compiled from the results of field research and the treatment of herbarium collections. The plant names of the species studied are given mainly according to vascular plants lists of the Far East [37,38,39,40,41,42,43,44,45].

Determining of the herbarium material was conducted in the laboratory of vegetation ecology at the Institute of Water and Ecology Problems, Far East Branch, Russian Academy of Science. The herbarium collected (over 700 sheets) is preserved in the Herbarium of the Institute of Water and Ecology Problems, Far East Branch, Russian Academy of Science (KHA). Standardized approaches were used for taxonomic analysis [36], and floristic reports were used to analyze flora by habitat type [37,38,39,40,41,42,43,44,45,46].

Separation of the core of the mire flora, or, in other words, species demonstrating the highest fidelity to mires, was conducted applying the J. Braun-Blanquet scale [20].

## Figures and Tables

**Figure 1 plants-11-00723-f001:**
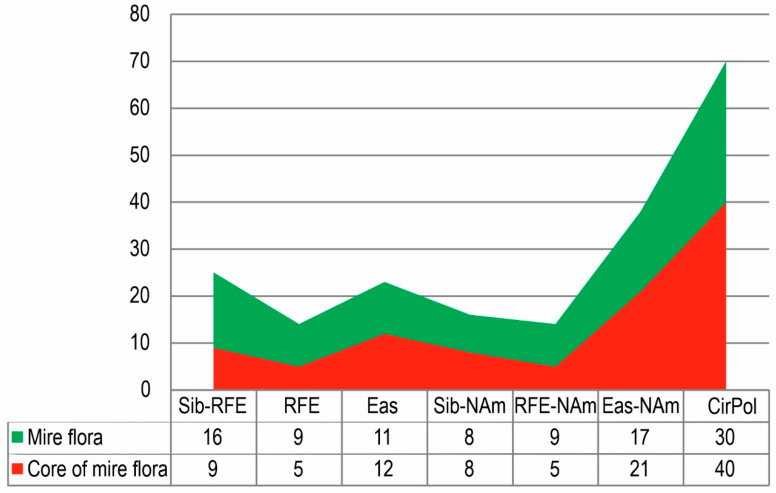
Percentage of longitudinal species groups in mire flora and its core of Bolshoy Shantar Island: Siberian-Russian Far Eastern (Sib-RFE), Russian Far Eastern (RFE), Eurasian (Eas), Siberian-North American (Sib-NAm), Russian Far Eastern-North American (RFE-NAm), Eurasian-North American (Eas-Nam), Circumpolar (CirPol).

**Figure 2 plants-11-00723-f002:**
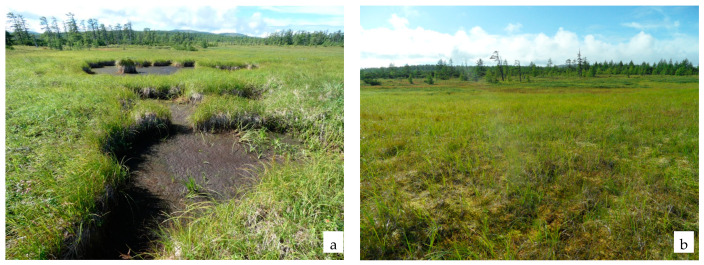
Sedge group of associations: northwestern coastal part of the Lake Bolshoe (**a**); estuary of Lake Bolshoe (**b**).

**Figure 3 plants-11-00723-f003:**
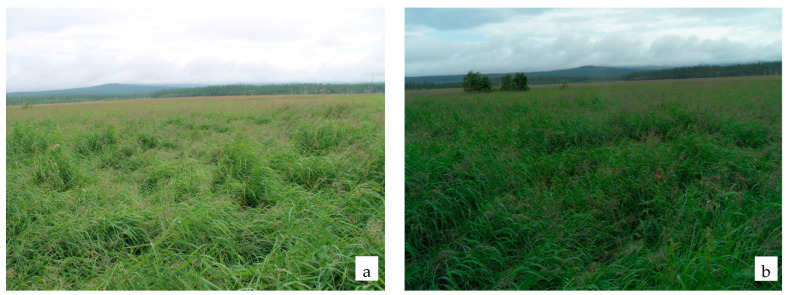
Reedgrass group of association: interstream area of the Argulad and the Levyi Argulad (**a**); estuary of Lake Bolshoe (**b**).

**Figure 4 plants-11-00723-f004:**
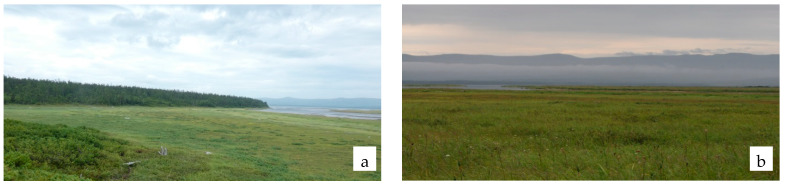
Coastal marshes: estuary of Lake Bolshoe (**a**), northwestern coastal part of the Lake Bolshoe (**b**).

**Figure 5 plants-11-00723-f005:**
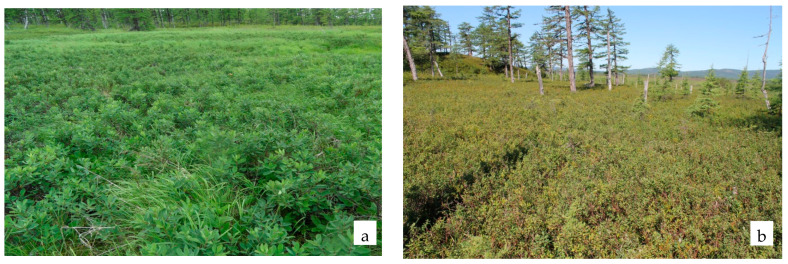
Shrub group of association: Northwestern coastal part of the Lake Bolshoe (**a**); interstream area of the Argulad and the Levyi Argulad (**b**).

**Figure 6 plants-11-00723-f006:**
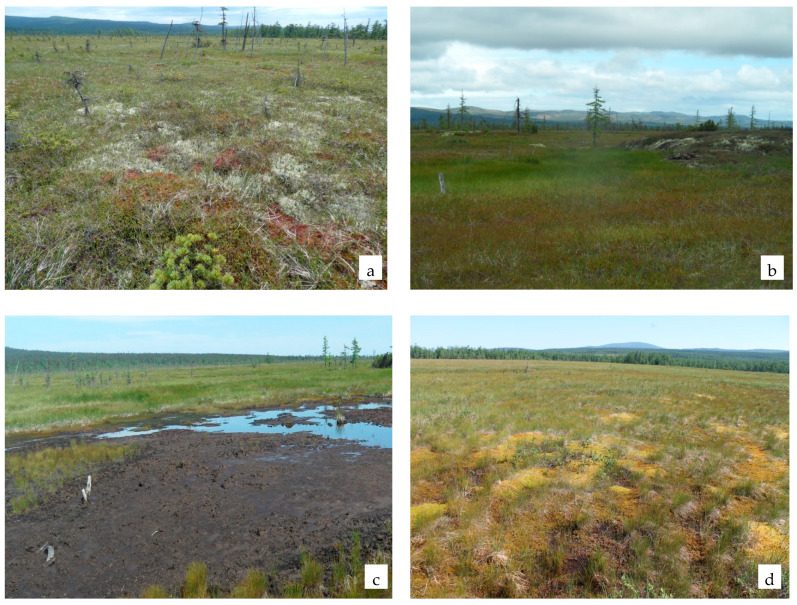
Sphagnum group of associations: interstream area of the Argulad and the Levyi Argulad (**a**,**b**); upper course of the Tundrovaya River (**c**,**d**).

**Figure 7 plants-11-00723-f007:**
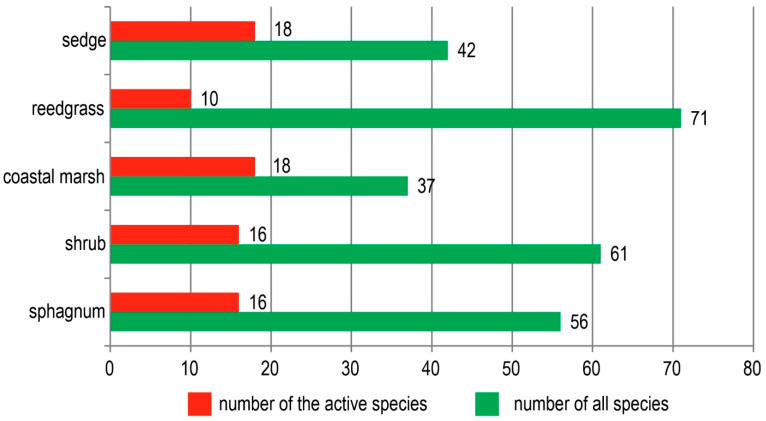
Percentage of total number of species and active species in five groups of associations.

**Figure 8 plants-11-00723-f008:**
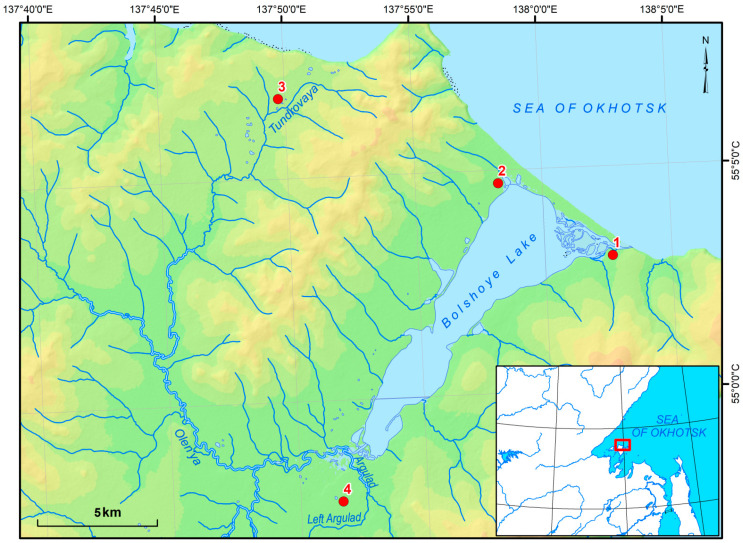
Research sites. 1. Estuary of Lake Bolshoe; 2. Northwestern coastal part of the Lake Bolshoe; 3. Upper course of the Tundrovaya River and 4. Interstream area of the Argulad and the Levyi Argulad.

**Table 1 plants-11-00723-t001:** Structure of head spectrum of flora of Shantar Archipelago, mire ecosystem of Bolshoy Shantar Island, and mire flora core of Bolshoy Shantar Island according to percentage of plant species in a family.

Vascular Plant Flora	Family Rank									
	1	2	3	4	5	6	7	8	9	10
					Rank 5–6				Rank 9–10	
The Shantar Archipelago	Po	Cy	Ra	As	Ro	Ca	Sa	Er	Pol	Sc
	10.1	9.1	8.6	8.0	4.7	4.7	4.6	2.9	2.7	2.7
Mire ecosystems of Bolshoy Shantar Island			Rank 3–5				Rank 7–8			
	Cy	Po	Er	Ra	Ro	Ca	As	Sc	Ap	Pol
	13.3	7.6	6.9	6.9	6.9	5.1	4.4	4.4	3.8	3.2
Core of mire flora of Bolshoy Shantar Island					Rank 5–10					
	Cy	Er	Ro	Ca	Po	Ra	Junc	Juncag	Dr	Pol
	32.4	9.2	7.7	4.6	3.1	3.1	3.1	3.1	3.1	3.1

Note: Po—*Poaceae*; Cy—*Cyperaceae*; Ra—*Ranunculaceae*; As—*Asteraceae*; Ro—*Rosaceae*; Ca—*Caryophyllaceae*; Sa—*Salicaceae*; Er—*Ericaceae*; Pol—*Polygonaceae*; Sc—*Scrophulariaceae*; Ap—*Apiaceae*; Junc—*Juncaceae*; Junag—*Juncaginaceae*; Dr—*Droseraceae*.

**Table 2 plants-11-00723-t002:** Topological-ecological classification of mire vegetation of Bolshoy Shantar Island.

Class	Group of Association	Association (Quantity)	Authors’ Description
Herbal euthrophic	Sedge	*Carex*-moss (4)	N10, N14, N28, U10
		*Carex*-*Myrica* (7)	N16, N18, N20, N34, U6, E7, A3
	Coastal marshes	Coastal marshes (5)	E1, E2, E3, E4, E2
	Reed grass	*Calamagrostis purpurea* (6)	E5, N7, N11, A26, A25, T17
		*Calamagrostis*-mixed herbs (4)	E11, N7, N11, N31
		*Calamagrostis-Carex* (2)	N1, N6
Wood- sphagnum eutrophic- mesotrophic	Shrub sphagnum with sparse larch	*Salix fuscescens*-*Myrica* (3)	N4, A18, A19
		*Salix fuscescens-Vaccinium uliginosum* (2)	N15, N30
		*Myrica* (4)	E8, N21, A17, N33
		*Salix fuscescens* (2)	N17, E9
		*Ledum*-*Vaccinium uliginosum-Sphagnum* with sparse larch (1)	A124
		*Vaccinium uliginosum-Carex-Sphagnum* with sparse larch (1)	A7
Sphagnum oligo- mesotrophic	Sphagnum	*Carex*-mixed herbs-*Sphagnum* (1)	T9
		Shrub-cotton grass-*Sphagnum* (1)	N19
		Dwarf shrub-*Carex*-*Sphagnum* (3)	T6, T7, T12
		Dwarf shrub-*Sphagnum* (5)	E13, T9, N13, N32, A11
		*Carex-Sphagnum* (3)	A1, A9, A10
Total associations: 54			

Note. In the authors’ description, number-letter indicates a profile: A—interstream area of the Argulad and the Levyi Argulad; E—estuary of Lake Bolshoe; T—Tundrovaya river; N—northwestern coastal part of the Lake Bolshoe.

**Table 3 plants-11-00723-t003:** Activity of vascular plants species of the sedge group of associations.

Mire Class/Group of Associations	Herbal Eutrophic/Sedge											
Authors’ Description Number	N10	N14	N16	N18	N20	N28	N34	E6	E7	E10	A3	Activity
*Myrica tomentosa*		5		5	20	1		10		20	5	III^+–2^
*Salix fuscescens*		5		15	5	1		40		30	15	III^+–3^
*Calamagrostis purpurea*	5	10	7	13	15	5	5	3	5	10	10	V^1–2^
*Sanguisorba tenuifolia*		1	1	2	2		1	2	2	2	1	V^1^
*Carex middendorfii*	20	5	10	15	10	20	60				40	IV^2–4^
*Arctanthemum arcticum*	1	1	3	1	1	1		1		2	1	IV^1^
*Comarum palustre*	5			5	1	1	3	3	3		3	IV^1^
*Carex cryptocarpa*	10		3	7	10	18		15	5			III^1−2^
*Chamaepericlymenum suecicum*		1	1	1	1					2		III^1^
*Drosera rotundifolia*		3		5	2				1	1		III^1^
*Oxycoccus palustris*		10		5	2				5	1		III^1–2^
*Iris setosa*			1	1	1			1	1			III^1^
*Parnassia palustris*			1	1	1				3	2		III^1^
*Carex rariflora*	10	10						20	15			II^2^
*Empetrum sibiricum*				3	3				15	10		II^1–2^
*Carex cinerea*								15				I^2^
*Carex globularis*											10	I^2^
*Carex loliacea*										15		I^2^

Note. Activity II^1^—*Galium trifidum* L., *Ligusticum scoticum* L., *Fimbripetalum radians*, *Equisetum pratense*, *Geranium erianthum*, *Luzula multiflora* (Ehrh. ex Retz.) Lej., *Menyanthes trifoliata* and *Hippuris lanceolata* Retz.; Activity I^1^—*Allium maximowiczii*, *Ligularia sibirica* (L.) Cass., *Baeothryon cespitosum*, *Equisetum fluviatile*, *Equisetum sylvaticum*, *Triglochin palustre*, *Epilobium palustre*, *Avenella flexuosa* (L.) Drejer, *Poa macrocalyx* Trautv. et Mey., *Rubus chamaemorus*, *Euphrasia ajanensis*, *Pedicularis resupinata* L. and *Gentiana glauca* Pall. The coverage of the species is shown by Arabic numerals according to the Braun-Blanquet scale [20] of species cover classes: +–<1%; 1—1–5%; 2—5–25%; 3—25–50%; 4—50–75%; 5—75–100%. Species constancy (with 20% class volume) classes are indicated by Roman numerals: I—1–20%, II—20–40%, III—40–60%, IV—60–80%, V—80–100% of areas. Empty cells indicate no species is present.

**Table 4 plants-11-00723-t004:** Activity of vascular plants species of the reedgrass group of associations.

Mire Class/Group of Associations	Herbal Eutrophic/Reedgrass												
Authors’ Description Number	E5	E11	N1	N6	N7	N8	N11	N29	N31	A26	A25	T17	Activity
*Calamagrostis purpurea*	80	30	50	20	90	10	20	20	70	60	20	21	V^2−5^
*Fimbripetalum radians*	1	1	3	1	1	2			2	2			IV^1^
*Sanguisorba tenuifolia*			5	4		3	2	3	2		1		III^1^
*Carex cryptocarpa*		8		5		5		2			4	4	III^1−2^
*Iris setosa*		1		2	1	3		5				1	III^1^
*Senecio cannabifolia*		5					6	1		1	3	3	III^1−2^
*Ranunculus repens*	1	1					2			1		1	III^1^
*Comarum palustre*		2	5						10	2		3	III^1−2^
*Carex middendorfii*			5	2		10			1				II^1−2^
*Leymus mollis*						5		10					II^1−2^

Note. Activity II^1^—*Larix cajanderi* Mayr, *Galium trifidum*, *Angelica saxatilis* Turcz. ex Ledeb., *Cicuta virosa* L., *Chamaepericlymenum suecicum*, *Equisetum pretense*, *Geranium erianthum*, *Veratrum lobelianum*, *Epilobium palustre*, *Parnassia palustris*, *Polemonium schmidtii*, *Tanacetum boreale* Fisch. ex DC. and *Galium boreale* L.; Activity I^1–2^—*Salix fuscescens*, *Salix pseudopentandra* (B. Floder.) B. Floder. and *Rosa rugosa*; Activity I^1^—*Caltha arctica* R.Br., *Allium maximowiczii*, *Ligusticum scoticum*, *Impatiens noli-tangere* L., *Cardamine regeliana* Miq., *Chamaepericlymenum canadense*, *Rhodiola integrifolia* Raf., C*arex globularis*, C*arex rhynchophysa* C. A. Mey., *Lathyrus pilosus*, *Luzula multiflora*, *Luzula rufescens* Fisch. ex E. Mey., *Bistorta vivipara* (L.) S. F. Gray, *Anemonidium dichotoma* (L.) Holub, *Thalictrum minus* L., *Thalictrum sparsiflorum* Turcz. ex Fisch. et Mey., *Rubus arcticus* L., *Sedum cyaneum* Rud., *Cirsium schantarense*, *Atriplex gmelinii* C. A. Mey., *Empetrum sibiricum*, *Equisetum fluviatile*, *Equisetum sylvaticum*, *Rhodococcum vitis-idaea*, *Juncus haenkei*, *Chamaenerion angustifolium*, *Poa pratensis* L., *Thalictrum contortum* L., *Filipendula palmate*, *Potentilla pacifica*, *Saxifraga aestivalis* Fisch. et Mey., *Pedicularis adunca* Bieb. ex Stev., *Urtica angustifolia*, *Gentiana glauca*, *Hippuris lanceolata*, *Artemisia leucophylla* (Turcz. ex Bess.) Clarke, *Rheum compactum* L., *Calamagrostis angustifolia* Kom., *Cacalia auriculata* DC., *Lonicera caerulea* L., *Lonicera chamissoi*, *Spiraea beauverdiana*, *Betula exilis* Sukacz., *Ledum palustre*, *Rosa acicularis* Lindl. and *Sorbus sambucifolia* Cham. et Schlecht. The coverage of the species is shown by Arabic numerals according to the Braun-Blanquet scale [20] of species cover classes: +–<1%; 1—1–5%; 2—5–25%; 3—25–50%; 4—50–75%; 5—75–100%. Species constancy (with 20% class volume) classes are indicated by Roman numerals: I—1–20%, II—20–40%, III—40–60%, IV—60–80%, V—80–100% of areas. Empty cells indicate no species is present.

**Table 5 plants-11-00723-t005:** Activity of vascular plants species of group of associations of coastal marshes.

Mire Class/Group of Associations	Herbal Eutrophic/Coastal Marshes					
Authors’ Description Number	E1	E2	E3	E4	N2	Activity
*Myrica tomentosa*			10		1	III^1^
*Salix fuscescens*			10		1	III^1^
*Potentilla pacifica*	20	40	10	50	5	V^2−3^
*Arctanthemum arcticum*	1	1	1	1		V^1^
*Stellaria humifusa*	2	2	3	2		V^1^
*Iris setosa*	1		1	1	1	V^1^
*Triglochin palustre*	40	10	15	2		V
*Allium maximowiczii*		2	1	2		IV^1^
*Arctopoa eminens*		20	1		1	IV^1−2^
*Carex loliacea*	2		3	15		IV^1−2^
*Lathyrus pilosus*	1	10	2			IV^1−2^
*Juncus haenkei*		15	3		2	IV^1−2^
*Calamagrostis purpurea*	10	10			25	IV^2^
*Leymus mollis*	1	18	8			IV^1−2^
*Chamaepericlymenum suecicum*			2		5	III^1^
*Geranium erianthum*			1		1	III^1^
*Calamagrostis langsdorffii*			3	10		III^1−2^
*Sanguisorba tenuifolia*			2		1	III^1^

Note: Activity I^1^—*Fimbripetalum radians*, *Atriplex gmelinii*, *Empetrum sibiricum*, *Juncus filiformis* L., *Luzula multiflora*, *Parnassia palustris*, *Calamagrostis deschampsioides* Trin., *Galium trifidum*, *Rubia jesoensis*, *Pedicularis adunca*, *Gentiana glauca*, *Sedum cyaneum*, *Carex* sp., *Calamagrostis tenuis* V. Vassil., *Tilingia ajanensis* Regel et Til., *Artemisia leucophylla*, *Poa palustris* and *Vicia cracca* L. The coverage of the species is shown by Arabic numerals according to the Braun-Blanquet scale [20] of species cover classes: +–<1%; 1—1–5%; 2—5–25%; 3—25–50%; 4—50–75%; 5—75–100%. Species constancy (with 20% class volume) classes are indicated by Roman numerals: I—1–20%, II—20–40%, III—40–60%, IV—60–80%, V—80–100% of areas. Empty cells indicate no species is present.

**Table 6 plants-11-00723-t006:** Activity of vascular plants species of the shrubs group of associations.

Mire Class/Group of Associations	Shrub-Eutrophic/Shrub													
Authors’ Description Number	N4	N15	N17	N21	N30	N33	E8	E9	E17	E18	E19	E14	E7	Activity
*Pinus pumila*	1			1		1							1	II^1^
*Myrica tomentosa*	10	5	2	20		5	20	1	60	40	40			IV^1−3^
*Salix fuscescens*	40	30	10	10	40	2	50	60		20	10			IV^1−4^
*Vaccinium uliginosum*	10	20	5		20	30		2				30	10	IV^1−3^
*Ledum palustre*						1		3				25		II^1−2^
*Spiraea beauverdiana*	10	3			2	2								II^1−2^
*Sanguisorba tenuifolia*	2	1	3	1	1	5	5	1	1	1	1			V^1^
*Carex middendorfii*	3	4	5	3	2	5			3	3	5		10	IV^1^
*Iris setosa*	1	1	2	1	1	3	1	1		1	1			IV^1^
*Empetrum sibiricum*	2	1	20	20		10	15			2	2		5	IV^1−2^
*Calamagrostis purpurea*	3	2	5		5	2			10	10	5			IV^1−2^
*Chamaepericlymenum suecicum*	1	1	1	1	3	1	2							III^1^
*Andromeda polifolia*			2	5						1	1	5	2	III^1^
*Parnassia palustris*	1	1					2		2	2	2			III^1^
*Carex cryptocarpa*		3		2	5			10				2		II^1−2^
*Carex globularis*									2	1	3	3	10	II^1−2^

Note: Activity II^1^—*Equisetum pretense*, *Lathyrus pilosus*, *Rubus chamaemorus*, *Arctanthemum arcticum*, *Carex rotundata*, *Drosera rotundifolia*, *Oxycoccus palustris*, *Geranium erianthum*, *Polemonium schmidtii*, *Allium maximowiczii*, *Stellaria longifolia* Muehl. ex Willd., *Comarum palustre*, *Galium trifidum* and *Pedicularis adunca*; Activity I^1–2^—B*aeothryon alpinum*, *Betula exilis* and *Calamagrostis langsdorffii*; Activity I^1^—*Larix cajanderi*, *Ligularia sibirica*, *Chamaedaphne calyculata*, *Oxycoccus microcarpus*, *Luzula multiflora*, *Platanthera tipuloides* (L. fil.) Lindl., *Avenella flexuosa*, *Angelica saxatilis*, *Cicuta virosa*, *Ligusticum scoticum*, *Maianthemum bifolium* (L.) F. W. Schmidt, *Tanacetum boreale*, *Lonicera chamissoi*, *Fimbripetalum radians*, *Chamaepericlymenum canadense*, *Carex rariflora*, E*riophorum russiolum*, *Equisetum fluviatile*, *Equisetum sylvaticum*, *Rhodococcum vitis-idaea*, *Hippuris tetraphylla*, P*halaroides arundinacea* (L.) Rausch., *Bistorta vivipara*, *Thalictrum sparsiflorum*, *Pedicularis lapponica* L., *Urtica angustifolia*, *Gentiana glauca* and *Menyanthes trifoliata*. The coverage of the species is shown by Arabic numerals according to the Braun-Blanquet scale [20] of species cover classes: +–<1%; 1—1–5%; 2—5–25%; 3—25–50%; 4—50–75%; 5—75–100%. Species constancy (with 20% class volume) classes are indicated by Roman numerals: I—1–20%, II—20–40%, III—40–60%, IV—60–80%, V—80–100% of areas. Empty cells indicate no species is present.

**Table 7 plants-11-00723-t007:** Activity of vascular plants species of the sphagnum group of associations.

Mire Class/Group of Associations	Sphagnum Oligo-Mesotrophic/Sphagnum													
Authors’ Description Number	T9	N19	T6	T7	T12	T13	T19	N13	N32	A1	A9	A10	A11	Activity
*Ledum palustre*		10	5	10	3	25	10	10	10	10	15	2	35	V^1−3^
*Vaccinium uliginosum*		10	5	10	3	30	10	10	10	5	10	2	35	V^1−3^
*Betula exilis*		2	2	2	1	1	2	5	3				1	IV^1^
*Myrica tomentosa*					20	5	40	3		20	5			III^1−3^
*Pinus pumila*		3	1		1			1	1				1	III^1^
*Carex globularis*	5		15	20	10	5	3	1	4	10	10	15	1	V^1−2^
*Empetrum sibiricum*	5	5	7	10	3	10	5	5	20	2	8	2		V^1−2^
*Rubus chamaemorus*		5	3	5	2		3	3	5	2	5	1	5	V^1^
*Carex middendorfii*		3	15	20	20	5		2	8	20	11	20	3	IV^1−2^
*Drosera rotundifolia*	2		2	2	2		2		2	1	2	2		IV^1^
*Andromeda polifolia*	1		2	2	2		2			2	5	2		IV^1^
*Oxycoccus microcarpus*		5	3		1		2	2	2	1			2	IV^1^
*Carex rotundata*			10	5	5		2			5	5	10		III^1−2^
*Chamaedaphne calyculata*		5		5	2	3				5		2	5	III^1^
*Eriophorum russiolum*		25				1				5	5	5	1	III^1−2^
*Oxycoccus palustris*	2	1	2	2	1							1		III^1^
*Carex cryptocarpa*	10	2												I^1−2^

Note: Activity II^1^—*Rhodococcum vitis-idaea*, *Platanthera tipuloides*, *Iris setosa*, *Parnassia palustris*, *Coptis trifolia* and *Baeothryon alpinum*; Activity I^1^—*Larix cajanderi*, *Carex rariflora*, *Equisetum vaginatum*, *Equisetum pretense*, *Calamagrostis purpurea*, *Bistorta vivipara*, *Sanguisorba tenuifolia*, *Allium maximowiczii*, *Seseli condensatum* (L.) Reichenb., *Maianthemum bifolium*, *Saussurea nuda* Ledeb., *Chamaepericlymenum suecicum*, *Baeothryon cespitosum*, *Carex loliacea*, *Rhynchospora alba* (L.) Vahl, *Equisetum fluviatile*, *Phyllodoce caerulea*, *Swertia tetrapetala*, *Geranium erianthum*, *Veratrum lobelianum*, *Gymnadenia conopsea*, *Polemonium schmidtii*, *Trientalis europaea* L., *Pyrola rotundifolia*, *Galium boreale*, *Pedicularis lapponica*, *Pedicularis labradorica* Wirsing, *Pedicularis resupinata* and *Gentiana glauca*. The coverage of the species is shown by Arabic numerals according to the Braun-Blanquet scale [20] of species cover classes: +–<1%; 1—1–5%; 2—5–25%; 3—25–50%; 4—50–75%; 5—75–100%. Species constancy (with 20% class volume) classes are indicated by Roman numerals: I—1–20%, II—20–40%, III—40–60%, IV—60–80%, V—80–100% of areas. Empty cells indicate no species is present.

## Data Availability

No new data were created or analyzed in this study. Data sharing is not applicable to this article.

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
