# Peer review of "Vascular Plants Flora of Mire Ecosystem of the Bolshoy Shantar Island (the Far East of Russia)"

_plants, 2022, doi:10.3390/plants11060723_

Round 1

Reviewer 1 Report

General: This manuscript presents a description of the vascular plant communities in the mire ecosystems of the Bolshoy Shantar island. This is a very interesting and remote island in the Russian Far East and so far little has been published in English about it. Furthermore, studies in the island could provide insight in evolutionary processes in the frame of island ecology, indirectly also argued by the authors. The study provides a very detailed descriptive overview of the botanical composition of such mire ecosystems. However, the study remains to be a population assessment and overview of the vascular plant species with hardly deeper analysis on underlying questions or the answering scientific hypotheses. Also, the English language seems to be not fully comply with the journal standards. I therefore recommend at the moment rather rejection of the current version. In case the authors can add there something more analytical with respective to their findings resubmission of a very much revised version should be possible. An improved version could be more suitable in another journal. The given information would be excellent for a book chapter for the respective region, but the manuscript is to my understandings currently lacking experimental design and analysis that goes beyond a pure description.

Introduction:

At the end the research objective is stated and this is very descriptive. Describing occurrence of plant species and their associations is of interest but I think you should include a higher more analytical level there. I think you should formulate here a research question to9 get a step further. E.g. how much does the composition of the respective associations that you describe differ form the main land? If so, this comparison needs to be quantified, so included additional methods. Or another research question of hypothesis in the context of your study. But you should go beyond pure description and listing of what you found. You have good data and a very good basis now for going further.

Material and Methods:

Very short, as the used methods are rather descriptive.

Results:

An interesting overview of the composition of different associations. Please add at each table a definition for used shortcuts (e.g. N10=…; N14= ….. etc.). Not all readers of the journal have a full botanical background. However, as mentioned earlier it remaisn to be purely descriptive and has hardly analytics to answer a certain question!

Discussion:

Too short I think. You gathered huge material and should more compare with information from similar regions or ecosystems and also include something about island and evolutionary aspects. As the island is situated in rather cool climate and close to the main land I expect not so much difference to plant communities on the neighbouring main land, right? However, you will have the better overview on this. But you should discuss here more of your findings in comparison with others.

Author Response

Dear reviewer!

We would like to express our deep gratitude for your attention to our article. Your comments are allowed us to improve the paper.

General: This manuscript presents a description of the vascular plant communities in the mire ecosystems of the Bolshoy Shantar island. This is a very interesting and remote island in the Russian Far East and so far little has been published in English about it. Furthermore, studies in the island could provide insight in evolutionary processes in the frame of island ecology, indirectly also argued by the authors. The study provides a very detailed descriptive overview of the botanical composition of such mire ecosystems. However, the study remains to be a population assessment and overview of the vascular plant species with hardly deeper analysis on underlying questions or the answering scientific hypotheses. Also, the English language seems to be not fully comply with the journal standards. I therefore recommend at the moment rather rejection of the current version. In case the authors can add there something more analytical with respective to their findings resubmission of a very much revised version should be possible. An improved version could be more suitable in another journal. The given information would be excellent for a book chapter for the respective region, but the manuscript is to my understandings currently lacking experimental design and analysis that goes beyond a pure description.

The authors added the comparative analysis with mire flora of mainland.

Introduction:

At the end the research objective is stated and this is very descriptive. Describing occurrence of plant species and their associations is of interest but I think you should include a higher more analytical level there. I think you should formulate here a research question to get a step further. E.g. how much does the composition of the respective associations that you describe differ from the main land? If so, this comparison needs to be quantified, so included additional methods. Or another research question of hypothesis in the context of your study. But you should go beyond pure description and listing of what you found. You have good data and a very good basis now for going further.Внесены дополнения.

The strategy of this scientific research was to establish and compare the zonal-regional and local patterns of mire vegetation of Bolshoy Shantar and marginal continental ecotone biogeosystems.

This article presents the outcomes of a study on the vascular plant flora of Bolshoy Shantar. The second stage of work involves a comparative analysis with the flora of the continental mire ecosystems of the coast of the Sea of Okhotsk. This will be done following similar studies in the adjacent continental territory. At present, this analysis is not possible because of the lack of data. No special studies have been conducted on the flora of vascular plants in the mire ecosystems of the Khabarovsk Krai. Information on species that grow in mires can be obtained from studies of flora and vegetation in northern Khabarovsk Krai [4,6,7–9]. They are shown in a general overview of species in large areas (Uda River basin, western coast of the Sea of Okhotsk, etc.), or incomplete lists of vegetation descriptions, and are insufficient to obtain reliable comparison results.

Material and Methods:

Very short, as the used methods are rather descriptive.

We added this section.

Results:

An interesting overview of the composition of different associations. Please add at each table a definition for used shortcuts (e.g. N10=…; N14= ….. etc.). Not all readers of the journal have a full botanical background. However, as mentioned earlier it remains to be purely descriptive and has hardly analytics to answer a certain question!

The authors gave the definition in each table. There are numbers of author’s descriptions for each group of association.

Discussion. Too short I think. You gathered huge material and should more compare with information from similar regions or ecosystems and also include something about island and evolutionary aspects. As the island is situated in rather cool climate and close to the main land I expect not so much difference to plant communities on the neighbouring main land, right? However, you will have the better overview on this. But you should discuss here more of your findings in comparison with others.

We added this section.

The mire ecosystems of the island are different from the coastal and continental ecosystems in that they have a large distribution and size of organogenic cryogenic landforms. They are much smaller on Bolshoy Shantar than on the mainland. The tree layer here is mainly represented by hypoarctic shrubs and shrubs and looks like tundra. The contrast of the vegetation of the insular ecosystems considered is most visibly reflected in regressive mire complexes, the nature of which is due to the characteristics of the coastal marine climate against the background of its warming in recent decades.

Our research has confirmed that there are 158 species of vascular plants in the flora of vascular plants in Bolshoy Shantar mires. Analysis of the literary sources dedicated to the vegetation in adjacent mainland areas has made it possible to include only 73 species in the similar mire ecosystems of the mainland, confirming the need for special studies on the flora of mires. When describing the mire vegetation, the authors of these papers usually take into account only the species that form the core of the mire flora. Species characterized by high persistence in mires, but capable of growing in other ecotopes, that are indifferent and random species to mire were not taken into account, but it is these species groups are indicators of the degree of transformation or the specificity of abiotic factors.

Nevertheless, preliminary comparative analysis can already reveal the specific features of the Bolshoy Shantar mire flora. It is defined by a group of plants that are genetically related to the Pacific coast. There are Rosa rugosa, Myrica tomentosa, Leymus mollis, Arctopoa eminens, and others. This group includes species of oceanic origin(Lysichiton camtschatcense, Rubia jesoensis, Lathyrus japonicus and etc.), that characterize the areal covering the mainland coast of the Sea of Okhotsk, Sakhalin, the Kuriles and Japan. Mountain species (Empetrum sibiricum, Arctous alpina, etc.) widely distributed in the Arctic and Eurasian continental mountains also play an important role in the formation of mire communities.

Therefore, the island effect that normally appear in flora depletion may not be represented by the flora of the mire ecosystems of Bolshoy Shantar, and conversely, the flora structure is complicated by the diversity of the floristic composition due to arctomontane and Pacific species. Due to the effects of the sea and the harsh climate, the flora structure of the island's mires has become more complex than that of the mires of the continental margins. At present, such comparative analysis is not possible due to the lack of complete data needed to compare the mainland and island floras of vascular plants in the ecosystems. This study aims to serve as a model that provides directions for future research in other parts of the Russian Far East.

We would hope that corrections made according your comment have led to substantial improvements of the manuscript.

Authors.

Reviewer 2 Report

The authors present an in-depth study of the mire ecosystem vascular plants of the island of Bolshoy Shantar.

The work is very detailed and refers to a very specific region.

Some tips for authors:

Introduction
- Line 36: "These forests expanded significantly as a result of fires related to whaling in the second half of the XX century". This sentence is equivocal

- Line 61: Eliminate the repetition "of the"

- Lines 78-83: This concept is the core of the work, perhaps it should also be summarized in the abstract.

Results
- Lines 119-125: It could be expressed in a more schematic way, for example with a bulleted list or with a table

- Table 1. Format better

Discussions
It would be appropriate to include considerations on the differences relating to different altitudes as well

Author Response

Dear reviewer!

Thank you very much for your comments. Please find below our responses to each of your comments.

Introduction

- Line 36: "These forests expanded significantly as a result of fires related to whaling in the second half of the XX century". This sentence is equivocal

Unlike many species in the  boreal zone which are susceptible to fire, larch are generally adapted to wildfires (Wirth 2005). Their thick bark can effectively prevent larch trees from being damaged by low-severity fires (Wirth 2005, Schulze et al 2012). They also drop low hanging branches, limiting the development of fuel ladders that facilitate the spread of fire into the forest canopy (Wirth 2005). In addition, larch forests are less prone to high-severity fires because their canopy closure is relatively low (Babintseva and Titova 1996, Kharuk et al 2010, Sofronov and Volokitina 2010, Kharuk et al 2011).

Wildfires are one of the major ecological processes in these forests. Recruitment of new larch seedlings is greatly aided by fires that consume much of the forests’ thick soil organic layer, which otherwise acts as a strong barrier to seedling establishment.

- Line 61: Eliminate the repetition "of the"

Corrected.

Lines 78-83: This concept is the core of the work, perhaps it should also be summarized in the abstract.

We agree with you, but the annotation is limited by 200 words.

Results

- Lines 119-125: It could be expressed in a more schematic way, for example with a bulleted list or with a table

- Table 1. Format better

Corrected.

Discussions

It would be appropriate to include considerations on the differences relating to different altitudes as well

It is very interesting offer. We are going to take into account it in the future work

Sincerely,

Authors

Round 2

Reviewer 1 Report

The manuscript has improved. However, there is still demand for improvements:

Introduction:

The research objectives should be stated more clear and concise. Currently there are a few sentences, not well formulated. This part has to be clear and well structured.

Materials and Methods:

This is now at the end of the Text, before the references? Why?

It should be before the results. The description this section, however, has improved.

Discussion:

Has improved and I found there the requested comparison with the continental mires. However, although not a native English speaker, I think that the English and the writing style needs improvements (overall manuscript and also especially this section).

Author Response

Dear reviewer!

We would like to thank you for your fruitful comments and suggestions. We tried to improve the manuscript according them.

Introduction:

The research objectives should be stated more clear and concise. Currently there are a few sentences, not well formulated. This part has to be clear and well structured.

Corrected.

Materials and Methods:

This is now at the end of the Text, before the references? Why?

It should be before the results. The description this section, however, has improved.

We followed the section order of our paper according to the requirements of journal.

Discussion:

Has improved and I found there the requested comparison with the continental mires. However, although not a native English speaker, I think that the English and the writing style needs improvements (overall manuscript and also especially this section).

Corrected.

With kind wishes,

Authors
